# Quantitative Analysis of Signal Heterogeneity in the Hepatobiliary Phase of Pretreatment Gadoxetic Acid-Enhanced MRI as a Prognostic Imaging Biomarker in Transarterial Chemoembolization for Intermediate-Stage Hepatocellular Carcinoma

**DOI:** 10.3390/cancers15041238

**Published:** 2023-02-15

**Authors:** Kiyoyuki Minamiguchi, Hideyuki Nishiofuku, Natsuhiko Saito, Takeshi Sato, Ryosuke Taiji, Takeshi Matsumoto, Shinsaku Maeda, Yuto Chanoki, Tetsuya Tachiiri, Hideki Kunichika, Takashi Inoue, Nagaaki Marugami, Toshihiro Tanaka

**Affiliations:** 1Department of Diagnostic and Interventional Radiology, Nara Medical University, Shijyocho 840, Kashihara City 634-8522, Japan; 2Department of Evidence-Based Medicine, Nara Medical University, Shijyocho 840, Kashihara City 634-8522, Japan

**Keywords:** hepatocellular carcinoma, Gadolinium ethoxybenzyl diethylenetriaminepentaacetic acid, magnetic resonance imaging, prognostic prediction, transarterial chemoembolization

## Abstract

**Simple Summary:**

Transarterial chemoembolization (TACE) is the recommended treatment for intermediate-stage hepatocellular carcinoma (HCC), and its role as a curative treatment is now greater than previously due to the introduction of targeted molecular and immunotherapies. An important current concern is patient selection in TACE for intermediate-stage HCC, which includes an extremely heterogeneous population. The hepatobiliary phase (HEB) in gadoxetic acid disodium-enhanced MRI (EOB-MRI) is related to β-catenin, so EOB-MRI could be a molecular imaging biomarker reflecting tumor biology. Although HCC with signal heterogeneity in the HBP of EOB-MRI showed malignant behavior, no previous study has evaluated prognosis after TACE based on signal heterogeneity quantified in the HBP of EOB-MRI. In this study, we showed a quantitative analysis of tumor signal heterogeneity in HBP on EOB-MRI was valuable in predicting the prognosis after TACE and suggested a treatment strategy for patients with intermediate-stage HCC based on this quantitative evaluation.

**Abstract:**

Background: In the era of local and systemic therapies for intermediate-stage hepatocellular carcinoma (HCC), personalized therapy has become available. The aim of our study was to evaluate the usefulness of quantitative analysis of pretreatment gadoxetic acid-enhanced magnetic resonance imaging (EOB-MRI) to predict prognosis following transarterial chemoembolization (TACE). Methods: This retrospective study included patients with treatment-naïve intermediate-stage HCC who underwent EOB-MRI before the initial TACE and were treated by initial TACE between February 2007 and January 2016. Signal heterogeneity in the hepatobiliary phase (HBP) of EOB-MRI was quantitatively evaluated by the coefficient of variation (CV). The cutoff CV value was determined using the Classification and Regression Tree algorithm. Results: A total of 64 patients were enrolled. In multivariate analysis, High CV (≥0.16) was significantly associated with poor prognosis (*p* = 0.038). In a subgroup analysis of patients within up-to-7 criteria, MST was significantly shorter in the High CV group than in the Low CV group (37.7 vs. 82.9 months, *p* = 0.024). In patients beyond up-to-7 criteria, MST was 18.0 and 38.3 months in the High CV and Low CV groups, respectively (*p* = 0.182). In both groups scanned at 1.5 T or 3.0 T, High CV was significantly associated with poor prognosis (*p* = 0.001 and 0.003, respectively). Conclusion: CV of the tumor in the HBP of EOB-MRI is a valuable prognostic factor of TACE.

## 1. Introduction

According to the latest update of the Barcelona Clinic Liver Cancer (BCLC) guidelines, trans-arterial chemoembolization (TACE) is still reserved and indicated as the first-line treatment for HCC patients at intermediate-stage (BCLC-B) who have well-defined nodules and can be treated by selective catheterization [1]. The role of TACE in the management of HCC in clinical practice is changing due to the recent development of targeted molecular and immunotherapies for HCC [2,3,4,5]. TACE now has a greater role as a curative treatment than previously [6,7,8,9].

An important current concern is patient selection in TACE for intermediate-stage HCC, which includes an extremely heterogeneous population. To stratify this heterogeneity, several subclassifications for intermediate-stage HCC have been suggested based on tumor burden and liver function [10,11,12,13]. In 2019, the concept of unsuitability for TACE was proposed at the Asia-Pacific Primary Liver Cancer Expert Meeting (APPLE) [14]. High tumor burden was one of the factors predictive of survival after TACE. Based on this system, “Beyond up-to-7 criteria” was adopted as “likely to develop TACE failure/refractoriness”. Other studies have recommended “up-to-11 criteria” as a more discriminative measure [11,15].

However, HCC is histologically and genetically diverse, and several subclassifications have been proposed based on clinical features, gene mutations, and biological pathways [16,17,18,19]. It is, therefore, important to evaluate the histological and genetic characteristics of tumors as well as the tumor burden in selecting personalized therapy.

Several previous reports have demonstrated that activation of β-catenin could be related to prognosis in HCC patients [16,17,18,19]. Activation of β-catenin is known to significantly correlate with the expression of organic anion transporting polypeptide 1B3 (OATP1B3) [20,21,22], which determines the uptake of gadoxetic acid disodium (Gd-EOB-DTPA). Therefore, Gd-EOB-DTPA-enhanced MRI(EOB-MRI) might be a molecular imaging biomarker predictive of patient prognosis. Although most HCCs are hypointense in the hepatobiliary phase (HBP) of EOB-MRI, it has been reported that approximately 6–15% of HCCs appear isointense or hyperintense due to overexpression of OATP1B3, which is termed paradoxical uptake [23]. Compared with hypointense tumors, those that exhibit hyperintensity in the HBP of EOB-MRI are generally considered to be less aggressive and to have a better prognosis [24,25,26,27]. However, this classification cannot be applied simply. In daily practice, heterogeneous intensity in HBP is found in many cases, including tumors that demonstrate combinations of isointense, hyperintense, and hypointense areas. Previously, Fujita et al. reported that HCC with signal heterogeneity in the HBP of EOB-MRI showed malignant behavior after surgical resection [28]. More recently, Lee et al. also reported that signal heterogeneity in the HBP of EOB-MRI was significantly associated with non-CR after drug-eluting bead (DEB)-TACE [29]. Their evaluation of EOB-MRI images was based on visual assessment.

The need for imaging parameters predictive of the response to TACE is of increased importance, particularly in the context of the increasingly widespread application of personalized medicine. However, to the best of our knowledge, no previous study has evaluated prognosis after TACE based on tumor heterogeneity quantified in the HBP of EOB-MRI.

The aim of our study was to evaluate the efficacy of quantitative analysis of EOB-MRI to predict prognosis after TACE in patients with intermediate-stage HCC.

## 2. Materials and Methods

### 2.1. Patients and Study Design

Our retrospective study was approved by the Institutional Review Board of Nara Medical University (No. 3119). Written informed consent was obtained from all patients before treatment. Included in this study were HCC patients classified as BCLC-B (intermediate-stage) who met the following inclusion criteria: (1) a diagnosis of HCC according to the guidelines of the Japan Society of Hepatology [30], (2) first TACE performed between February 2007 and January 2016, (3) Child–Pugh A or B liver function (Child–Pugh score 5–7), and (4) EOB-MRI performed before the initial TACE treatment. Patients who had additional treatment within 4 weeks after TACE (radiofrequency ablation [RFA], surgery, radiation therapy, or molecular targeted agents) were excluded.

### 2.2. Transarterial Chemoembolization (TACE) Procedure

In all patients, digital subtraction angiography (DSA), CT during hepatic arteriography (CTHA), and CT during arterial portography (CTAP) were performed prior to TACE for evaluation of vascular anatomy, tumor vascularity, tumor number, tumor location, and portal vein patency. To detect tumor-feeding arteries, maximum intensity projection (MIP) images were generated using a three-dimensional (3D) CT workstation (Ziostation; Ziosoft, Tokyo, Japan (February 2007–August 2008) and Synapse Vincent; Fujifilm, Tokyo, Japan (September 2008–January 2016)). The tumor-feeding arteries were selectively catheterized using a 1.5–2.0-Fr tip microcatheter with reference to this navigation image. An emulsion was prepared by manually mixing ethiodized oil (Lipiodol Ultra-Fluide; Guerbet, Villepinte, France) with an anti-cancer drug (epirubicin in most cases, cisplatin in some cases) with a three-way stopcock. The amount of emulsion was determined based on tumor diameter, tumor number, and liver function. The maximum doses of ethiodized oil and drugs used were as follows: ethiodized oil, 10 mL; epirubicin, 60 mg; and cisplatin, 100 mg. TACE was performed by injecting the emulsion followed by gelatin sponge particles (Gelpart; Nippon Kayaku, Tokyo, Japan; diameter, 1 mm). The endpoint of TACE was disappearance of tumor staining on DSA. In the case of difficulty in superselective TACE, such as multiple tumors, a microcatheter was inserted as close to the tumor as possible to avoid non-target embolization. In the case of tumor size >6 cm, bland transarterial embolization (TAE) using gelatin sponge particles (Gelpart) or microspheres (Embosphere; Merit Medical, South Jordan, UT, USA) was initially performed to reduce tumor volume and decrease tumor vascularity. Following bland TAE, conventional TACE was performed as mentioned above. In several patients aged >80 years, epirubicin-loaded drug-eluting-bead TACE was performed using DC Beads (Boston Scientific, Marlborough, MA, USA). TACE with cisplatin plus gelatin particles was carried out in patients enrolled in a clinical trial.

### 2.3. Gadoxetic Acid-Enhanced MRI and Image Analysis

MRI examinations in our institution were performed using a 1.5 T (MAGNETOM Avanto or MAGNETOM Symphony Sonata; Siemens Healthineers, Erlangen, Germany) or a 3.0 T (MAGNETOM Skyra or MAGNETOM Verio) system. Although the 1.5 T scanner was utilized at other institutions, information regarding the scanner was unavailable due to its limitations. EOB-MRI was performed prior to TACE (median time before TACE, 34.5 days; range, 1–267 days). Pre- and post-contrast 3D fat-suppressed T1-weighted images were acquired using the following protocols. Avanto: TR/TE = 3.69/1.37 ms, flip angle 12°, field of view 380 mm, matrix 256 × 173, and slice thickness 5 mm. Sonata: TR/TE = 3.45/1.41 ms, flip angle 12°, field of view 350 mm, matrix 320 × 187, and slice thickness 3 mm. Skyra: TR/TE = 3.77/1.45 ms, flip angle 10°, field of view 350 mm, matrix 448 × 329, and slice thickness 3 mm. Verio: TR/TE = 3.61/1.39 ms, flip angle 10°, field of view 350 mm, matrix 480 × 257, and slice thickness 3 mm. Gd-EOB-DTPA (Primovist; Bayer Healthcare, Berlin, Germany) was injected intravenously at a dose of 0.1 mL/kg (0.025 mmol/kg) at a rate of 1.5 mL/s, followed by a saline flush of 30 mL. Arterial phase imaging was acquired immediately after the contrast agent reached the right ventricle. Five arterial phases were acquired in a single breath hold (approximately 20–45 s after injection of EOB). Ten seconds later, two portal phases were obtained, each with 20 s breath hold. Delayed phase was defined as approximately 3 min after injection, and HBP images were obtained 20 min after injection.

Two radiologists specializing in abdominal imaging (T.T., H.N.) measured tumor signal intensity in the HBP images using regions of interest (ROIs). For each nodule identified in the HBP, an ROI was placed manually on the 2D slice that showed the largest tumor diameter (Figure 1). Any disagreements about ROI placement were solved in consensus. The coefficient of variation (CV) was calculated for each tumor as the standard deviation (SD, σ) divided by the mean signal intensity (μ), as an indicator of the relative variability of voxels within the ROI. The mean CV of the two readers was used in this study. CV was calculated using the following formula:(1)$$\mathrm{coefficient}\;\mathrm{of}\;\mathrm{variation}\;(\mathrm{CV})=\frac{\sigma }{\mu }.$$

### 2.4. Evaluation of Cutoff for Coefficient of Variation

The Classification and Regression Tree (CART) was employed to evaluate the best CV cutoff threshold predictive of overall survival time [31,32]. The patients were divided into two groups (High CV and Low CV groups) according to the CV cutoff value identified by CART.

### 2.5. Follow-Up and Evaluation

Contrast-enhanced CT (CECT) was performed at 1–3 months after TACE to evaluate response based on the modified Response Evaluation Criteria In Solid Tumors (mRECIST) [33]. Tumor response was evaluated by a radiologist specialized in abdominal imaging (K.M.) [34]. Patients were followed up by CECT every 3 months until recurrence of HCC occurred. Follow-up laboratory data were also obtained, including tumor markers, to evaluate HCC recurrence. In the case that CT could not detect recurrence despite elevated tumor markers, repeat EOB-MRI was performed. Repeat TACE was performed if local or new intrahepatic recurrence was detected.

Overall survival was defined as the length of time from the date of first TACE until the date of death. Prothrombin time activation, albumin, total bilirubin, α-fetoprotein (AFP), patient age, background liver disease, interval between the first and second TACE sessions, and sequential treatment after TACE were reviewed based on clinical and laboratory records. Tumor number and tumor size were evaluated from the radiological records by two radiologists (T.T., H.N.).

### 2.6. Statistical Analysis

All statistical analyses were performed using IBM SPSS statics version 26 (SPSS, Inc., Chicago, IL, USA). Inter-reader agreement was evaluated using the intraclass correlation coefficient (ICC) and classified into five categories: 0.0–0.20 as poor; 0.21–0.40 as fair; 0.41–0.60 as moderate; 0.61–0.80 as good; and 0.81–1.00 as excellent. Overall survival was assessed using the Kaplan–Meier method. The log-rank test was used to compare survival curves between the two groups. Univariate and multivariate analyses were performed using Cox proportional hazard models to analyze the prognostic factors of TACE. Candidates for multivariate analysis were those with values of *p* < 0.05 in the univariate analysis. Clinical factors were compared between the groups using Chi-squared test or Mann–Whitney U test. *p* values < 0.05 were considered statistically significant.

## 3. Results

### 3.1. Patients

A total of 91 patients satisfied the inclusion criteria. Of these, 27 patients who had additional treatment within 4 weeks after TACE were excluded from this study. Thus, a total of 64 patients with 188 nodules were enrolled in this study: 47 men (73.4%) and 17 women (26.6%), with a median age of 74 years (range 50–89 years). The background liver disease was viral hepatitis B or C in 49 patients (76.6%) and non-viral infection in 9 patients (14.1%). There were 59 patients (92.2%) with a Child–Pugh score of 5 or 6, and 5 patients with a Child–Pugh score of 7 (7.8%). The largest tumor diameter ranged from 1.5 to 14.0 cm, and the mean diameter was 4.4 ± 2.66 cm. Of the total patients, 85.9% received conventional TACE, and 3.1% received DEB-TACE. The average number of TACE procedures per patient was 3.5 (range, 1–10). In 1.5 T, 26, 7, and 8 patients were scanned utilizing the Avanto, Sonata, and scanners from other institutions. In 3.0 T, 1 and 22 patients were scanned utilizing the Skyra and Verio scanners, respectively. The baseline patient characteristics are shown in Table 1.

### 3.2. Treatment Outcome

After excluding 2 patients who did not undergo contrast-enhanced CT, 62 patients were evaluated for tumor response. The objective response rate (ORR) was 93.5% (58/62). Complete response (CR), defined as the disappearance of any intratumoral arterial enhancement in all target lesions, was achieved in 43 patients (69.4%), partial response in 15 patients (24.2%), and stable disease in 4 patients (6.5%). No patient had progressive disease. Median OS was 46.7 months (95%CI, 30.5–62.9). The cumulative OS rates at 1, 2, and 3 years were 89.9%, 75.9%, and 62.7%, respectively (Figure 2).

### 3.3. CV Cutoff Value

The ICC for CV between the two readers was 0.97 (95% confidence interval, 0.95–0.98). The optimal cutoff value of CV used for patient classification was 0.16, determined by the CART procedure based on survival time analysis of the event of mortality within 2 years. In this study, clinical features other than CV were not added to the CART algorithm since CV was the most excellent independent variable to classify the target subjects into two groups with the cutoffs, according to the log-rank test in the Kaplan–Meier method and hazard ratio in the Cox proportional hazard model (Appendix A). Patients with at least one nodule of *CV* ≥0.16 (*n* = 21, 32.8%) were classified into the High CV group. Those in which all nodules had CV values <0.16 (n = 43, 67.2%) were classified into the Low CV group.

### 3.4. Univariate and Multivariate Analyses of Prognostic Factors

Table 2 lists the results of univariate and multivariate Cox regression analyses performed to evaluate the prognostic factors of TACE. Among 13 prognostic factors, univariate analysis revealed 6 independent factors to be significantly correlated with unfavorable prognosis: serum AFP >200 ng/mL (*p* = 0.005), Child–Pugh score of 7 (*p* = 0.016), ALBI grade 2 (*p* = 0.02), beyond up-to-7 (*p* = 0.002), beyond up-to-11 (*p* = 0.042), and High CV (*p* = 0.001). To prevent multicollinearity, multivariate analysis was performed using CV, up-to-7, AFP, and the Child–Pugh score. In multivariate analysis, High CV was significantly associated with poor prognosis (*p* = 0.038).

### 3.5. Overall Survival and CR Rate between the High CV and Low CV Groups

The CR rate after the initial TACE was lower in the High CV group (55.0%) than in the Low CV group (76.2%), but did not reach statistical significance (*p* = 0.091). There was a significant difference between the groups in terms of the median interval between the first and second TACE sessions (High CV group, 3.2 months; Low CV group, 8.9 months; *p* = 0.013). The 1-, 3-, and 5-year survival rates were 95.1%, 74.3%, and 48.3%, respectively, in the Low CV group, and 78.2%, 35.9%, and 14.4%, respectively, in the High CV group. MST was 32.4 months and 57.3 months in the High CV and Low CV groups, respectively (*p* < 0.001) (Figure 3).

### 3.6. Subgroup Analysis by Up-To-7 Criteria

Of patients within up-to-7 criteria, MST was significantly shorter in the High CV group than in the Low CV group (37.7 months vs. 82.9 months, *p* = 0.024). The 1-, 3-, and 5-year survival rates were 100%, 79.3%, and 57.4%, respectively, in the Low CV group; and 75%, 56.3%, and 18.8%, respectively, in the High CV group (Figure 4a). Of patients beyond up-to-7 criteria, MST was shorter in the High CV group, but the difference was not significant (18.0 months vs. 38.3 months, *p* = 0.182). The 1-, 3-, and 5-year survival rates were 81.8%, 62.3%, and 24.9%, respectively, in the Low CV group; and 80%, 20%, and 10%, respectively, in the High CV group (Table 3) (Figure 4b).

### 3.7. Overall Survival between the High CV and Low CV Groups, Respectively, Scanned at 1.5 T and 3.0 T

The optimal CV cutoff value determined by the CART procedure for patients scanned at 1.5 T and 3.0 T was 0.102 and 0.167, with high CV in 31 and 7 patients and low CV in 10 and 16 patients, respectively. MST was 30.0 months in the High CV and not reached in Low CV groups scanned at 1.5 T (*p* = 0.001). MST was 21.6 months and 53.1 months in the High CV and Low CV groups scanned at 3.0 T, respectively (*p* = 0.003). (Figure 5).

## 4. Discussion

Several studies have attempted to predict prognosis in HCC using the signal intensity of EOB-MRI. Some demonstrated that HCCs with signal heterogeneity in the HBP of EOB-MRI showed more malignant behavior compared with other HCCs [28,29,35,36]. Although the reason for greater malignant potential in HCCs with signal heterogeneity in the HBP remains unclear, Fujita et al. proposed that the signal intensity of HCC in HBP changes from homogeneous hypointensity or hyperintensity to heterogeneous hyperintensity with increasing degree of malignancy [28].

Tumor signal intensity in the HBP of EOB-MRI is determined by the uptake of Gd-EOB-DTPA via OATP1B3, an uptake transporter [37,38]. It is known that the expression of OATP1B3 correlates with β-catenin activation [20]. Several reports have shown that β-catenin mutation was associated with a relatively favorable prognosis in HCC [16,17,18,19]. Among β-catenin activated HCCs, those with the expression of hepatocyte nuclear factor (HNF) 4α are reported to have a good prognosis and show iso-to-high signal intensity in the HBP of EOB-MRI. However, the prognosis in β-catenin activated HCCs is controversial. Some previous studies have reported that β-catenin mutation induced epithelial–mesenchymal transformation (EMT) of epithelial cells to the highly aggressive phenotype [39,40,41]. Xu et al. have demonstrated that HCC with increased β-catenin expression showed poorer overall survival and progression-free survival (PFS) than HCC with negative β-catenin expression [42]. Kitao et al. also reported a group of patients who had poor prognoses among those with β-catenin mutation in HCC [27]. HCC with heterogeneous intensity in the HBP might be related to β-catenin mutation, which results in poor prognosis due to EMT.

To date, no study has quantified the heterogeneity of tumor signal intensity in the HBP of EOB-MRI. CV is a relative measure of variability that enables the assessment of heterogeneity and has been used as a parameter for the evaluation of signal heterogeneity in various regions [43,44]. Recent reports have evaluated the usefulness of CV for predicting malignancy and therapeutic effects in other cancers [45,46]. High CV corresponds to the high heterogeneity of signal intensity in the HBP. The present CV cutoff value was calculated based on a survival period of 2 years, according to the MST of 20.4–27.6 months for intermediate-stage HCC previously reported for beyond up-to-7 criteria [47,48].

Previous studies revealed that high tumor burdens, such as beyond up-to-7 criteria or up-to-11 criteria, were poor prognostic factors in TACE for intermediate-stage HCC [11,14,15]. Both beyond up-to-7 criteria and beyond up-to-11 criteria were revealed as significant prognostic factors in the present univariate analysis. However, multivariate analysis identified only High CV as a significant prognostic factor (*p* = 0.038). This finding suggests that the evaluation of malignant potential on EOB-MRI could be more important than tumor size or number in this regard. Surprisingly, even within up-to-7 criteria, MST was significantly shorter for High CV than for Low CV.

The heterogeneity of tumor signal intensity on the HBP of EOB-MRI has been reported to significantly correlate with non-CR after DEB-TACE [29]. We found no significant correlation between CV and objective response after TACE (*p* = 0.091), although the CR rate was lower in the High CV group than in the Low CV group (55.0% vs. 76.2%). The latest BCLC guidelines state that TACE is the first-line treatment for intermediate-stage HCC, but do not specify which type of TACE should be performed. To date, there is no evidence of the superiority of conventional TACE or DEB-TACE for intermediate-stage HCC [49,50,51]. We mainly used conventional TACE in the patients enrolled in the present study. It is necessary to perform a further examination of objective response in a larger population due to our small study population.

The optimal CV cutoff value scanned by 3.0 T was higher than that of 1.5 T. The CV, defined as the ratio of the SD to the mean, is inversely proportional to the signal-to-noise ratio, which is associated with the field strength. The field strength is expected to affect the CV value. Subgroup analysis by up-to-7 criteria according to each field strength is difficult due to the small sample size of this study, and further research evaluation is needed.

On the basis of the present results, the following treatment strategy could be considered (Table 4). (i) For patients with Low CV/within up-to-7 (MST of 82.9 months in our study), TACE alone can be used. (ii) For patients with High CV/within up-to-7 and Low CV/beyond up-to-7 (MST of 37.7 and 38.3 months, respectively, in our study), the combination of TACE and systemic therapy is recommended. (iii) For patients with High CV/beyond up-to-7 (MST of 18 months in our study), current molecular-targeted agents or immunotherapy should be considered first. A previous study reported that MST was 37.9 months in patients with beyond up-to-7 criteria who were treated with initial lenvatinib with or without additional TACE [52].

Several recent studies have reported that EOB-MRI could be a useful imaging biomarker for drug selection in advanced HCC [53,54,55,56,57,58]. HCC with a high enhancement ratio in the HBP of EOB-MRI showed resistance to immunotherapy due to β-catenin mutation [53,54]. Therefore, High CV groups, including those with partial uptake of Gd-EOB-DTPA, could also show resistance to immunotherapy. Our proposed CV might be helpful not only for identifying patients suitable for TACE, but also for the selection of personalized systemic therapy.

Our study has several limitations. First, this single-center retrospective study included a small number of subjects, and it is, therefore, necessary to validate the obtained cutoff value in an external validation cohort. Second, due to the retrospective nature of this study, there may have been changes in the state of the art in terms of imaging techniques and TACE procedures. Third, sequential treatment after the initial TACE varied. Fourth, we did not obtain the immunohistochemistry findings, i.e., β-catenin and HNF4α. Fifth, parameters were assessed in images obtained with scanners of different magnetic field strengths (1.5 T and 3.0 T). Finally, we evaluated CV on a single slice with the maximum tumor diameter. Therefore, further investigations are required, including assessment using 3D volumes of interest.

## 5. Conclusions

In conclusion, quantitative analysis of signal heterogeneity in the HBP of EOB-MRI using CV is useful for predicting prognosis following TACE. This technique could be helpful in determining treatment strategies for patients with intermediate-stage HCC. Even in patients within up-to-7 criteria, those with High CV could be considered for combined treatment with TACE and systemic therapy.

## Figures and Tables

**Figure 1 cancers-15-01238-f001:**
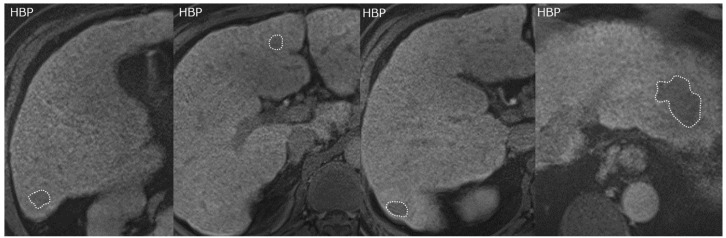
A representative case of HCC classified as Low CV/beyond up-to-7. An ROI is placed on each tumor in the HBP of EOB-MRI (dotted regions). Of the six tumors in this patient, four representative examples are shown. All tumors showed hyperenhancement in the arterial phase and hypointensity in the HBP of EOB-MRI. Maximum tumor diameter is 35.5 mm, and maximum CV is 0.138. OS, defined as the time from the date of initial treatment to death by any cause, was 46.7 months in this patient. CV, coefficient of variation; EOB-MRI, gadoxetic acid-enhanced magnetic resonance imaging; HBP, hepatobiliary phase; HCC, hepatocellular carcinoma; OS, overall survival; ROI, region of interest.

**Figure 2 cancers-15-01238-f002:**
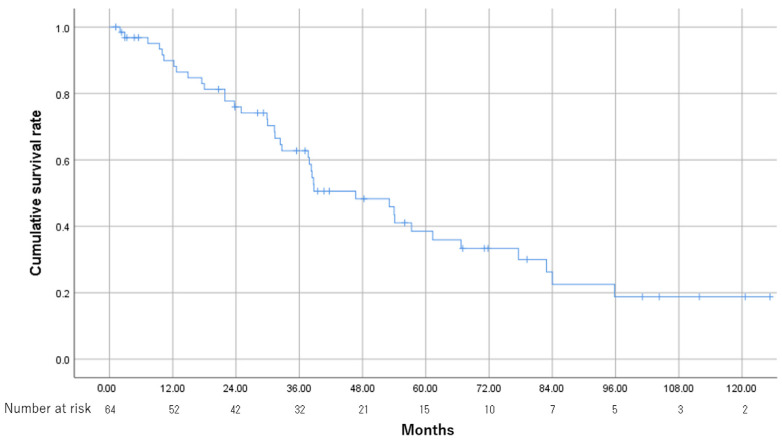
Kaplan–Meier analysis of OS in 64 patients with intermediate-stage hepatocellular carcinoma treated with TACE. The overall survival rates at 1, 2, and 3 years were 89.9%, 75.9%, and 62.7%, respectively. OS, overall survival; TACE, transarterial chemoembolization. RFA was performed in 17 patients, surgery in 2 patients, and arterial infusion chemotherapy in 19 patients as sequential treatment after downstaging by TACE. Molecular-targeted therapy was used for 12 patients (18.8%) who became refractory to TACE.

**Figure 3 cancers-15-01238-f003:**
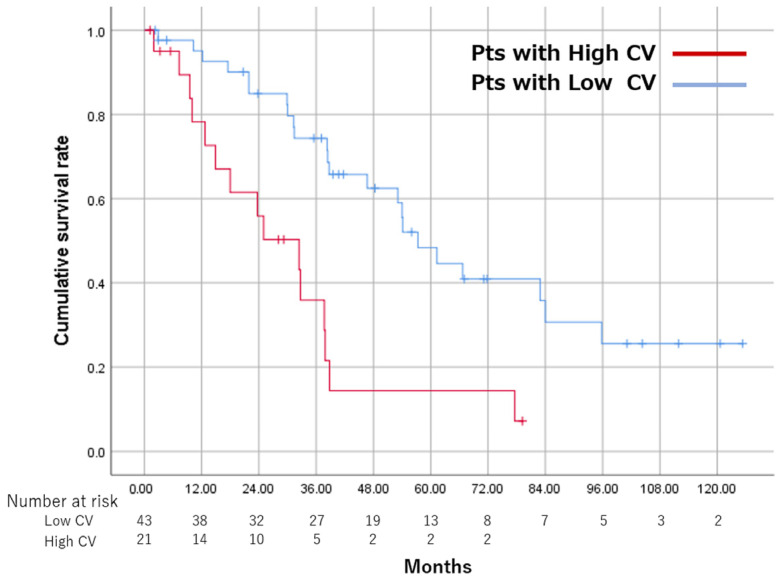
Comparison of OS in 64 patients with intermediate-stage hepatocellular carcinoma treated with TACE, between two groups divided based on a cutoff value of CV 0.16. CV, coefficient of variation; OS, overall survival; Pts, patients; TACE, transarterial chemoembolization.

**Figure 4 cancers-15-01238-f004:**
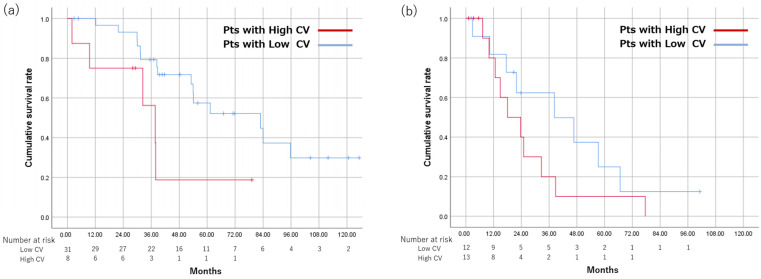
Comparison of OS using a cutoff value of CV 0.16 among patients divided based on up-to-7 criteria. OS is shown in patients within up-to-7 criteria (**a**) and beyond up-to-7 criteria (**b**). CV, coefficient of variation; OS, overall survival; Pts, patients.

**Figure 5 cancers-15-01238-f005:**
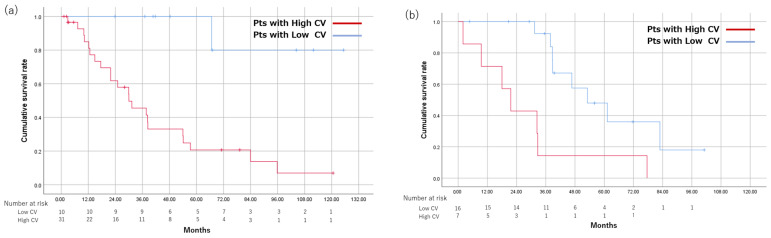
Comparison of OS with each CV cutoff value in patients divided by scanner type. OS is shown in patients scanned at 1.5 T (**a**) and 3.0 T (**b**). CV, coefficient of variation; OS, overall survival; Pts, patients.

**Table 1 cancers-15-01238-t001:** Clinical features of 64 patients with intermediate-stage hepatocellular carcinoma.

Characteristic	All Patients (*n* = 64)	%
Age (year)		
<80	58	90.6
≥80	6	9.4
Sex		
Male	47	73.4
Female	17	26.6
Child-Pugh score		
5.6	59	92.2
7	5	7.8
PT (%)		
<70	3	4.7
≥70	61	95.3
Total bilirubin(mg/dL)		
<2	63	98.4
≥2	1	1.6
Albumin(g/dL)		
<3.5	3	4.7
≥3.5	61	95.3
AFP (ng/mL)		
<200	54	84.4
≥200	10	15.6
Up-to-7		
in	39	60.9
out	25	39.1
Up-to-11		
in	57	89.1
out	7	10.9
Etiology of liver disease		
No	6	9.4
Yes	58	90.6
ALBI grade		
1	23	35.9
2	41	64.1
MTA after refractory to TACE		
No	52	81.2
Yes	12	18.8
MRI scanner		
1.5 T		
Avanto	26	40.6
Sonata	7	10.9
Other institutions	8	12.5
3.0 T		
Skyra	1	1.6
Verio	22	34.4

Note: AFP, α-fetoprotein; ALBI, albumin-bilirubin; MTA, molecular targeted agent; PT, prothrombin time; TACE, transarterial chemoembolization.

**Table 2 cancers-15-01238-t002:** Univariate and multivariate analysis of risk factors associated with overall survival among patients with intermediate-stage hepatocellular carcinoma.

	Univariate Analysis		Multivariate Analysis	
Risk Factor	*p*-Value	HR (95% CI)	*p*-Value	HR (95% CI)
Age ≥ 80 years	0.339	2.049 (0.472–8.903)		
Etiology of liver disease	0.364	0.575 (0.174–1.901)		
Coefficient of variation ≥0.16	0.001	3.211 (1.615–6.384)	0.038	2.354 (1.049–5.281)
AFP ≥ 200 ng/mL	0.005	3.208 (1.428–7.207)	0.673	1.244 (0.452–3.429)
Up-to-7 out	0.002	2.745 (1.440–5.236)	0.157	1.777 (0.801–3.943)
Up-to-11 out	0.042	3.810 (1.048–13.852)		
Child–Pugh score 7	0.016	3.688 (1.270–10.704)	0.190	2.145 (0.685–6.719)
ALBI grade 2	0.02	2.663 (1.163–6.096)		
PT < 70%	0.262	1.986 (0.599–6.588)		
Totalbilirubin ≥ 2.0 mg/dL	0.528	0.047 (0.000–621.057)		
Albumin < 3.5 g/dL	0.072	2.031 (0.938–4.397)		
Post-TACE MTA	0.071	1.886 (0.947–3.755)		
Non-CR after first TACE	0.072	0.525 (0.260–1.060)		

Note: AFP, α-fetoprotein; ALBI, albumin-bilirubin; MTA, molecular targeted agent; PT, prothrombin time; TACE, transarterial chemoembolization; HR, hazard ratio; CI, confidence interval.

**Table 3 cancers-15-01238-t003:** Subgroup analysis of up-to-7 criteria.

	Within Up-To-7 (*n* = 39)		Beyond Up-To-7 (*n* = 25)	
	High CV (*n* = 8)	Low CV (*n*=31)	High CV (*n* = 13)	Low CV (*n* = 12)
MST (months)	37.7	82.9	18	38.3

Note: CV, coefficient of variation; MST, median survival time.

**Table 4 cancers-15-01238-t004:** Treatment strategy according to CV level and tumor burden.

	Within Up-To-7	Beyond Up-To-7
Low CV	TACE	TACE + systemic therapy
High CV	TACE + systemic therapy	Systemic therapy

Note: CV, coefficient of variation; TACE, transarterial chemoembolization.

## Data Availability

All data generated or analyzed during this study are included in this article. Further inquiries can be directed to the corresponding author.

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
