# Peer review of "Quantitative Analysis of Signal Heterogeneity in the Hepatobiliary Phase of Pretreatment Gadoxetic Acid-Enhanced MRI as a Prognostic Imaging Biomarker in Transarterial Chemoembolization for Intermediate-Stage Hepatocellular Carcinoma"

_cancers, 2023, doi:10.3390/cancers15041238_

Round 1

Reviewer 1 Report

The manuscript entitled "Quantitative Analysis of Signal Heterogeneity in the Hepatobiliary Phase of Pretreatment Gadoxetic Acid-enhanced MRI as a Prognostic Imaging Biomarker in Transarterial Chemoembolization for Intermediate-stage Hepatocellular Carcinoma" presents a study which quantitatively evaluated the the signal heterogeneity of hepatocellular carcinoma (HCC) in the hepatobiliary phase of gadoxetic acid-enhanced MRI (EOB-MRI) to predict prognosis after TACE treatment. This study retrospectively enrolled 91 patients. After excluding 27 patients who had additional treatment within 4 weeks after TACE, 64 patients were included in this study. The MRI data were acquired from four different scanners at 1.5T or 3.0T. Two radiologists independently drew a ROI on a 2D slice for each tumor, the coefficient of variation (CV) was calculated for each HCC tumor for analysis. The classification and regression tree (CART) algorithm was used to evaluate the best cutoff CV threshold which separates high and low CV groups. The results showed that the high CV group had lower survival rate and median survival time than low CV group.

Overall, the study utilized CV metric to quantitatively measure the signal inhomogeneity within the tumors, but the 

heterogeneity of imaging data and small sample size may lead to errors in the results. Although the study showed some interesting findings, I have some comments to the manuscript.

Major issues

1) Because of retrospective study design, the imaging data were acquired from patients using both 1.5T and 3.0T MR scanners. However, it has been demonstrated that field strength is associated with signal-to-noise ratio (SNR) and contrast-to-noise ratio (CNR), and higher field strength has higher SNR and CNR in the acquired images. Because  the CV is defined as (standard deviation)/(mean value) which is inversely related to SNR, it is strongly suggested to use imaging data acquired on the same field strength. 

2) Different scanners and different imaging parameters are also influential factors that can lead to errors in the results because imaging parameters are also associated with SNR and CNR in the acquired images. So, it is  recommended to use imaging data acquired with identical imaging parameters on the same scanner.

3) Authors should clearly provide information about the distribution of patient imaging data acquired from different scanners in Table 1.

4) Authors utilized machine learning to separate imaging data with low and high CV. The information about separating the training and testing sets should be described in details. Did author try other machine learning models? 

Minor issues

1) On line 163, OS should be defined at first use.

2) On lines 167-170, the details about building the machine learning model should be described, such as separation of training and testing sets. How is the accuracy and stability of the model?

3) On line 249, CR should be defined at first use.

Author Response

Dear Reviewer1:

Thank you very much for the review of our manuscript entitled: “Quantitative Analysis of Signal Heterogeneity in the Hepatobiliary Phase of Pretreatment Gadoxetic Acid-enhanced MRI as a Prognostic Imaging Biomarker in Transarterial Chemoembolization for Intermediate-stage Hepatocellular Carcinoma”. Our reply to the reviewer's comments is provided below. We have also made appropriate corrections as indicated by the reviewers.

[Major issues]

Comment 1

Because of retrospective study design, the imaging data were acquired from patients using both 1.5T and 3.0T MR scanners. However, it has been demonstrated that field strength is associated with signal-to-noise ratio (SNR) and contrast-to-noise ratio (CNR), and higher field strength has higher SNR and CNR in the acquired images. Because the CV is defined as (standard deviation)/(mean value) which is inversely related to SNR, it is strongly suggested to use imaging data acquired on the same field strength. 

Reply 1
Thank you very much for reviewing our manuscript. We sincerely appreciate all valuable suggestions, which helped us to improve the quality of our article.
According to your valuable comments, we conducted Kaplan-Meier survival analysis using optimal CV cutoff value determined by the CART procedure for patients scanned at 1.5T and 3.0T respectively.

We added Figure.5 and the following sentence in Results;

3.7. Overall Survival between the High CV and Low CV groups, respectively, scanned at 1.5T and 3.0T

The optimal CV cutoff value determined by the CART procedure for patients scanned at 1.5T and 3.0T was 0.102 and 0.167, with high CV in 31 and 7 patients and low CV in 10 and 16 patients, respectively. MST was 30.0 months in the High CV and not reached in Low CV groups scanned at 1.5T (P =0.001). MST was 21.6 months and 53.1 months in the High CV and Low CV groups scanned at 3.0T, respectively (P =0.003). (Figure 5)” (Lines.291-298)

However, unfortunately, subgroup analysis by up-to-7 criteria according to each field strength is difficult due to the small sample size of this study. Please see the below figures. We attached the result of subgroup analysis by up-to-7 criteria according to each field strength.

And we added the following sentence in Discussion;

“The optimal CV cutoff value scanned by 3.0T was higher than that of 1.5T. The CV, defined as the ratio of the SD to the mean, is inversely proportional to the signal-to-noise ratio, which is associated with the field strength. The field strength is expected to affect the CV value. Subgroup analysis by up-to-7 criteria according to each field strength is difficult due to the small sample size of this study, and further research evaluation is needed.” (Lines.354-359)

Comment 2

Different scanners and different imaging parameters are also influential factors that can lead to errors in the results because imaging parameters are also associated with SNR and CNR in the acquired images. So, it is recommended to use imaging data acquired with identical imaging parameters on the same scanner.

 Reply 2

Thank you very much for your important comments.

According to your valuable comments, we calculated the optimal CV cutoff value by the CART procedure for each of 4 scanners. However, the number of cases scanned by Sonata and Skyra was too small to obtain the CV cutoff value. As you mentioned, we consider it important to evaluate CV with identical imaging parameters on the same scanner, and this needs to be evaluated further in the future.

Comment 3

Authors should clearly provide information about the distribution of patient imaging data acquired from different scanners in Table 1.

 Reply 3

Thank you very much for this suggestion.

We added the distribution of patient for each scanner to the Table 1.

And we added the following sentence in Results;

“In 1.5 T, 26, 7, and 8 patients were scanned utilizing the Avanto, Sonata, and scanners from other institutions. In 3.0 T, 1 and 22 patients were scanned utilizing the Skyra and Verio scanners, respectively.” (Lines.214-216)

Comment 4

Authors utilized machine learning to separate imaging data with low and high CV. The information about separating the training and testing sets should be described in details. Did author try other machine learning models? 

Reply 4

Thank you very much for these valuable suggestions.

We did not use any machine learning techniques. Classification and Regression Trees (CART) method is a nonparametric decision tree learning technique that generates either classification or regression trees, depending on whether the dependent variable is categorical or numeric. The CART which uses historical data based on survival analysis to construct decision trees is a useful method for estimating appropriate cut-off values when predicting time-to-event event rates for continuous variables such as biomarkers. So, we used this method just for estimating an appropriate CV cut-off value in this study. We did not utilize machine learning to separate imaging data with low and high CV. To avoid misunderstanding, we have changed the following sentence in the Materials and Methods:

From: “The Classification and Regression Tree (CART), a predictive algorithm used in machine learning, was employed to evaluate the best CV cutoff threshold predictive of overall survival time [31, 32]”

To: “The Classification and Regression Tree (CART) was employed to evaluate the best CV cutoff threshold predictive of overall survival time [31, 32]” (Lines.170-171)

[Minor issues]

Comment 1

On line 163, OS should be defined at first use.

Reply 1

Thank you for your valuable comments.

We changed the sentences as follows;

From; “OS was 46.7 months in this patient.”

To; “OS, defined as the time from the date of initial treatment to death by any cause, was 46.7 months in this patient.” (Line.165)

Comment 2

On lines 167-170, the details about building the machine learning model should be described, such as separation of training and testing sets. How is the accuracy and stability of the model?

Reply 2

Thank you very much for your important suggestions.

As we used the CART just for estimating appropriate CV cut-off values in this study, we did not utilize any machine learning model to separate imaging data with low and high CV. 

In general, the validity of the classification by the CART is assessed by the mean and its standard deviation of the misclassification using the cross-validation method. However, in this study, the validity and its stability were confirmed by the difference in survival rate and the number at risk between the two groups separated on the Kaplan Meier diagram.

Comment 3

On line 249, CR should be defined at first use.

Reply 3

Thank you for your valuable comments.

We changed the sentences as follows;

From; “Complete response (CR) was achieved in 43 patients (69.4%), partial response in 15 patients (24.2%), and stable disease in 4 patients (6.5%).”

To; “Complete response (CR), defined as disappearance of any intratumoral arterial enhancement in all target lesions, was achieved in 43 patients (69.4%), partial response in 15 patients (24.2%), and stable disease in 4 patients (6.5%).” (Lines.226-227)

Reviewer 2 Report

As a result of the analysis of the article developed by your team, we found that it is included among the articles that have the treatment of hepatocellular carcinoma as the subject of study. Furthermore, the article addresses the treatment by the TACE method of hepatocellular carcinoma in the intermediate stage and proposes the quantitatively evaluated coefficient of variation of the signal heterogeneity from the hepatobiliary phase of EOB-MRI as a prognostic factor of TACE.

I find that the selection of patients was carried out according to a well-established protocol, the number of patients included in the study is sufficient to obtain statistical significance, the statistical analysis is adequate and the results are expressed in graphs and tables in which the parameters selected to be studied can be identified .

The discussions are carried out in comparison with the results of other studies that had as a study subject the treatment of hepatocellular carcinoma by the TACE method; studies that are highlighted in the recent bibliography related to the article.

The conclusion of the study is pertinent and is supported by the results and discussions in the article

Author Response

Dear Reviewer2:

Thank you very much for the review of our manuscript entitled: “Quantitative Analysis of Signal Heterogeneity in the Hepatobiliary Phase of Pretreatment Gadoxetic Acid-enhanced MRI as a Prognostic Imaging Biomarker in Transarterial Chemoembolization for Intermediate-stage Hepatocellular Carcinoma”. Our reply to the reviewer's comments is provided below.

Comment

As a result of the analysis of the article developed by your team, we found that it is included among the articles that have the treatment of hepatocellular carcinoma as the subject of study. Furthermore, the article addresses the treatment by the TACE method of hepatocellular carcinoma in the intermediate stage and proposes the quantitatively evaluated coefficient of variation of the signal heterogeneity from the hepatobiliary phase of EOB-MRI as a prognostic factor of TACE.

I find that the selection of patients was carried out according to a well-established protocol, the number of patients included in the study is sufficient to obtain statistical significance, the statistical analysis is adequate and the results are expressed in graphs and tables in which the parameters selected to be studied can be identified.

The discussions are carried out in comparison with the results of other studies that had as a study subject the treatment of hepatocellular carcinoma by the TACE method; studies that are highlighted in the recent bibliography related to the article.

The conclusion of the study is pertinent and is supported by the results and discussions in the article

Reply
Thank you very much for taking the time to review our manuscript.

We really appreciate it.

Round 2

Reviewer 1 Report

In the revised manuscript, I have some comments:

1) Authors determined the optimal CVs for 1.5T and 3.0T, respectively, which I think was a suitable approach to avoid the effect of field strength. Authors should add relevant results in the abstract.

2) The main concern was in the CART algorithm. Because the patients' data enrolled were heterogeneous in age, sex, AFP, etc., which may be associated with survival time, it is not clear whether those clinical features were added in the CART algorithm to find the optimal CV cutoff values. 

3) If clinical features were added in the algorithm, it is important to demonstrate that in addition to clinical features, CV is an independent variable for predicting survival rate/time.

Author Response

Point by point reply

Reviewer1. 

Comment 1

Authors determined the optimal CVs for 1.5T and 3.0T, respectively, which I think was a suitable approach to avoid the effect of field strength. Authors should add relevant results in the abstract.

Reply 1
Thank you for your valuable comments.
We added the following sentence in the Abstract.

“In both groups scanned at 1.5T or 3.0T, High CV was significantly associated with poor prognosis (P=0.001 and 0.003, respectively).” (Lines.43-44)

Comment 2

The main concern was in the CART algorithm. Because the patients' data enrolled were heterogeneous in age, sex, AFP, etc., which may be associated with survival time, it is not clear whether those clinical features were added in the CART algorithm to find the optimal CV cutoff values. 

 Reply 2

Thank you very much for your important comments.

In this study, clinical features other than CV (sex, age, AFP etc) were not added in the CART algorithm, because CV was the most excellent independent variable to classify the target subjects into two groups (higher and lower risk) with the cutoffs, according to the log-rank test in the Kaplan-Meier method and HR in the Cox proportional hazard model.

At first, we show a univariate Cox proportional hazards analysis with the observation period up to 2 years, using life or death after 2 years as a response variable (Supplementary Table.S1). CV had the highest HR as an explanatory variable but was not significant for P-value. This may be due to the inclusion of both 1.5T and 3T magnetic fields.

We also show the Kaplan-Meier method with CV classified by each magnetic field, in addition to Up-to-7 and PT, which had significant P-values, as explanatory variables (Supplementary Fig.S1-S2). In the group scanned by 3.0T, the analysis using PT as an explanatory variable was difficult since there was no case of PT<70%.

As a result, the most excellent explanatory variable to classify the target subjects into high and low risk groups was CV.

Comment 3

If clinical features were added in the algorithm, it is important to demonstrate that in addition to clinical features, CV is an independent variable for predicting survival rate/time.

 Reply 3

Thank you very much for your valuable suggestions.

CV was the most excellent independent variable to classify the target subjects into two groups (higher and lower risk) with the cutoffs, according to the log-rank test in the Kaplan-Meier method and HR in the Cox proportional hazard model as we showed in reply 2.

We added the following sentence in the Result.

“In this study, clinical features other than CV were not added to the CART algorithm since CV was the most excellent independent variable to classify the target subjects into two groups with the cutoffs, according to the log-rank test in the Kaplan-Meier method and hazard ratio in the Cox proportional hazard model (Supplementary Figure S1-S2, Table S1).” (Lines.246-250)
